# Unsupervised machine learning predicts future sexual behaviour and sexually transmitted infections among HIV-positive men who have sex with men

**Sara Andresen**[1,2☯]*, **Suraj Balakrishna**[1,2☯], **Catrina Mugglin**[3], **Axel J. Schmidt**[4,5], **Dominique L. Braun**[1,2], **Alex Marzel**[6], **Thanh Doco Lecompte**[7], **Katharine EA Darling**[8], **Jan A. Roth**[9,10,11], **Patrick Schmid**[4], **Enos Bernasconi**[12], **Huldrych F. Günthard**[1,2], **Andri Rauch**[3‡], **Roger D. Kouyos**[1,2‡], **Luisa Salazar-Vizcaya**[3‡], **the Swiss HIV Cohort Study**

1 Division of Infectious Diseases and Hospital Epidemiology, University Hospital Zurich, Zurich, Switzerland, 2 Institute of Medical Virology, University of Zurich, Zurich, Switzerland, 3 Department of Infectious Diseases, Bern University Hospital Inselspital, University of Bern, Bern, Switzerland, 4 Division of Infectious Diseases and Hospital Epidemiology, Cantonal Hospital St. Gallen, St. Gallen, Switzerland, 5 Sigma Research, London School of Hygiene and Tropical Medicine, United Kingdom, 6 Research, Teaching and Development, Schulthess Clinic, Zurich, Switzerland, 7 HIV Unit, Infectious Diseases Division, Department of Medicine, University Hospital of Geneva, Switzerland, 8 Infectious Diseases Service, Department of Medicine, University Hospital of Lausanne (CHUV), Switzerland, 9 Division of Infectious Diseases and Hospital Epidemiology, University Hospital Basel, Basel, Switzerland, 10 Basel Institute for Clinical Epidemiology and Biostatistics, University Hospital Basel, University of Basel, Basel, Switzerland, 11 Division of Analytical and Research Services, Department of Informatics, University Hospital Basel, Basel, Switzerland, 12 Division of Infectious Diseases, Lugano Regional Hospital, Lugano, Switzerland

☯ These authors contributed equally to this work.
‡ AR, RDK, and LS-V also contributed equally to this work.
* sara.andresen@bluewin.ch

**Data Availability Statement:** The data underlying the study cannot be openly shared for ethical and legal reasons, as they contain potentially identifying

## Abstract

Machine learning is increasingly introduced into medical fields, yet there is limited evidence for its benefit over more commonly used statistical methods in epidemiological studies. We introduce an unsupervised machine learning framework for longitudinal features and evaluate it using sexual behaviour data from the last 20 years from over 3'700 participants in the Swiss HIV Cohort Study (SHCS). We use hierarchical clustering to find subgroups of men who have sex with men in the SHCS with similar sexual behaviour up to May 2017, and apply regression to test whether these clusters enhance predictions of sexual behaviour or sexually transmitted diseases (STIs) after May 2017 beyond what can be predicted with conventional parameters. We find that behavioural clusters enhance model performance according to likelihood ratio test, Akaike information criterion and area under the receiver operator characteristic curve for all outcomes studied, and according to Bayesian information criterion for five out of ten outcomes, with particularly good performance for predicting future sexual behaviour and recurrent STIs. We thus assess a methodology that can be used as an alternative means for creating exposure categories from longitudinal data in epidemiological models, and can contribute to the understanding of time-varying risk factors.

and sensitive information, but interested researchers can apply to access the data: 1) The SHCS informed consent states that sharing data outside the SHCS network is only permitted for specific studies on HIV infection and its complications, and to researchers who have signed an agreement detailing the use of the data and biological samples; and 2) the data is too dense and comprehensive to preserve patient privacy in persons living with HIV. According to the Swiss law, data cannot be shared if data subjects have not agreed or data is too sensitive to share. Investigators with a request for selected data should send a proposal to the respective SHCS address (www.shcs.ch/contact). The provision of data will be considered by the Scientific Board of the SHCS and the study team and is subject to Swiss legal and ethical regulations and is outlined in a material and data transfer agreement. However, all codes are available on https://github.com/Kouyos-Group/Behavioural-Clusters-and-STIs.

**Funding:** This study has been financed within the framework of the Swiss HIV Cohort Study, supported by the Swiss National Science Foundation (grant 201369), by the SNF project grant 324730_179567, and by the SHCS research foundation (project number 823). RDK and SB were supported by the Swiss National Science Foundation (grant BSSGI0_155851). The funders had no role in study design, data collection and analysis, decision to publish, or preparation of the manuscript.

**Competing interests:** I have read the journal's policy and the authors of this manuscript have the following competing interests: DLB reports honoraria and travel grants outside of the submitted work from Gilead, ViiV and Merck. HFG, outside of this study, reports grants from Swiss HIV Cohort Study, grants from Swiss National Science Foundation, during the conduct of the study; grants from Swiss HIV Cohort Study, grants from Swiss National Science Foundation, grants from NIH, grants from Gilead unrestricted research grant, personal fees from Advisor/consultant for Merck, ViiV healthcare and Gilead sciences and member of DSMB for Merck, grants from Yvonne Jacob Foundation. KEAD's institution has received research funding unrelated to this publication from Gilead and sponsorship to specialist meetings from MSD. AR reports fees for sitting on advisory boards from Merck Sharp & Dohme and Gilead Sciences; travel grants from Gilead Sciences, Pfizer, and AbbVie; and a research grant from Gilead Sciences, outside of the submitted work. All

## Author summary

Machine learning tools are becoming increasingly important in medical research. Yet it is not entirely clear whether these tools perform better than conventional ones when it comes to research on a population level. Here we designed and tested a machine learning method which we use to predict sexually transmitted diseases among HIV-positive men who have sex with men in Switzerland. We used a machine learning algorithm to find groups of men with similar sexual behaviour in the last twenty years, and found that considering these groups in addition to conventional risk factors yielded more accurate predictions of who would be diagnosed with an STI. With this we hope to shed some light on how and to what extent machine learning can help predict and prevent infectious diseases.

## Background

In recent years, machine learning methods have been increasingly introduced into medical research and practice [1,2]. While pattern recognition algorithms are commonly associated with imaging or laboratory diagnostics, their benefit over conventional statistics is more uncertain in epidemiological analyses [1]. Here we introduce an unsupervised machine learning framework for longitudinal data, and evaluate it using sexual behaviour data to predict future sexual behaviour and STIs.

Sexually transmitted infections (STIs) remain a clinical and research priority worldwide. Despite major advances in treatment and prevention, the incidence of STI diagnoses, especially among men who have sex with men (MSM), remains high and has increased in several countries [3–7]. For instance, syphilis incidence among HIV-positive MSM in the Swiss HIV Cohort Study (SHCS) increased from 30.1 to 59.2 per 1000 patient-years between 2006 and 2017, with high rates of repeated syphilis episodes [8,9]. Meanwhile, hepatitis C virus infections most likely associated with unprotected anal intercourse and sexualised injection drug use increased from 0.23 to 4.09 per 100 patient-years between 1998 and 2011 in the same cohort [6].

Behavioural factors have been repeatedly identified as drivers for these STI epidemics, particularly multiple sexual partners and sex without condoms [6,7,10,11]. And, importantly, several studies have shown increases in condomless anal intercourse and number of sexual partners among MSM in recent years [12–14]. To develop effective treatment and prevention programs for STIs, it is crucial to understand sexual behaviour patterns and their role in STI transmission. However, human behaviour is extremely heterogeneous and challenging to capture [15]. Further, time-varying variables often enter statistical models as binary values, such as "ever [had sex without a condom]" or "recently [had sex without a condom]". However, there is a wealth of information in the trajectory of a time series, though this data is often very high-dimensional and hence challenging to incorporate in statistical models.

We have previously shown that in HIV-positive MSM in the SHCS unsupervised machine learning reveals subgroups of this population with distinct behavioural patterns over time [16]. For this, clusters of individuals were inferred based on their history of condom use for anal intercourse with non-steady partners over the past two decades. This clustering algorithm can be seen as a means to distil information from these time series into a single variable (in this case, cluster membership). What remains unclear is whether or how these behavioural clusters are relevant to future sexual behaviour and STI acquisition.

fees were paid to AR's institution and not to AR personally.

In this work, we thus propose a framework to validate whether clusters inferred by unsupervised machine learning can enhance predictions of sexual behaviour, incident physician- or study nurse-reported STIs and incident laboratory-confirmed syphilis among HIV-positive MSM in the near future. To further characterise the heterogeneity of sexual behaviour among MSM in the SHCS, we examine whether behavioural clusters are associated with participants' number of sexual partners.

## Methods

### Ethics statement

The SHCS and the ZPHI have been approved by the ethics committees of the participating institutions (Ethikkommission beider Basel ("Die Ethikkommission beider Basel hat die Dokumente zur Studie zustimmend zur Kenntnis genommen und genehmigt."); Kantonale Ethikkommission Bern (21/88); Comité départemental d'éthique des spécialités médicales et de médecine communautaire et de premier recours, Hôpitaux Universitaires de Genève (01–142); Commission cantonale d'éthique de la recherche sur l'être humain, Canton de Vaud (131/01); Comitato etico cantonale, Repubblica e Cantone Ticino (CE 813); Ethikkommission des Kan-tons St. Gallen (EKSG 12/003); Kantonale Ethikkommission Zürich (KEK-ZH-NR: EK-793 for the SHCS and EK-1452 for the ZPHI)) and written informed consent has been obtained from all participants.

### Swiss HIV cohort study and Zurich Primary HIV infection study

The SHCS is an ongoing nationwide prospective cohort that routinely collects behavioural, laboratory, and clinical data from HIV-positive persons in Switzerland since 1988 (www.shcs.ch) [17]. Individual data are recorded at enrolment and every six months thereafter. It has been estimated that more than 80% of all MSM currently diagnosed with HIV in Switzerland are followed in the cohort [18].

The Zurich Primary HIV Infection study (ZPHI) is an ongoing monocenter cohort [19]. Between 2015 and 2016, participants completed a detailed sexual behaviour and STI symptom questionnaire in addition to their SHCS follow-up (all ZPHI participants included in this study are also enrolled in the SHCS) [11].

### Inference of behavioural clusters

Clusters of participants with similar sexual behaviour over time were inferred based on the method described in [16]. In brief, at SHCS follow-up every six months, each participant is asked the following questions:

a. "Have you had sex with non-steady partners in the last six months?"; if yes

b. "Did you have anal or vaginal intercourse with these partners?"; if yes,

c. "Did you use condoms all the time?"

Participants who replied "yes" to question (a) where considered to have sex with non-steady partners (nsP), and those who replied "yes" to (a) and (b) and "no" to (c) were considered to have condomless anal intercourse with non-steady partners (nsCAI). Based on these variables, a trajectory was constructed for each participant recording their nsCAI or nsP "state" at each timepoint (Fig 1, Step 1).

Behavioural clusters were inferred based on these sexual behaviour trajectories: First, the similarity between trajectories was assessed for each pair of participants by computing the

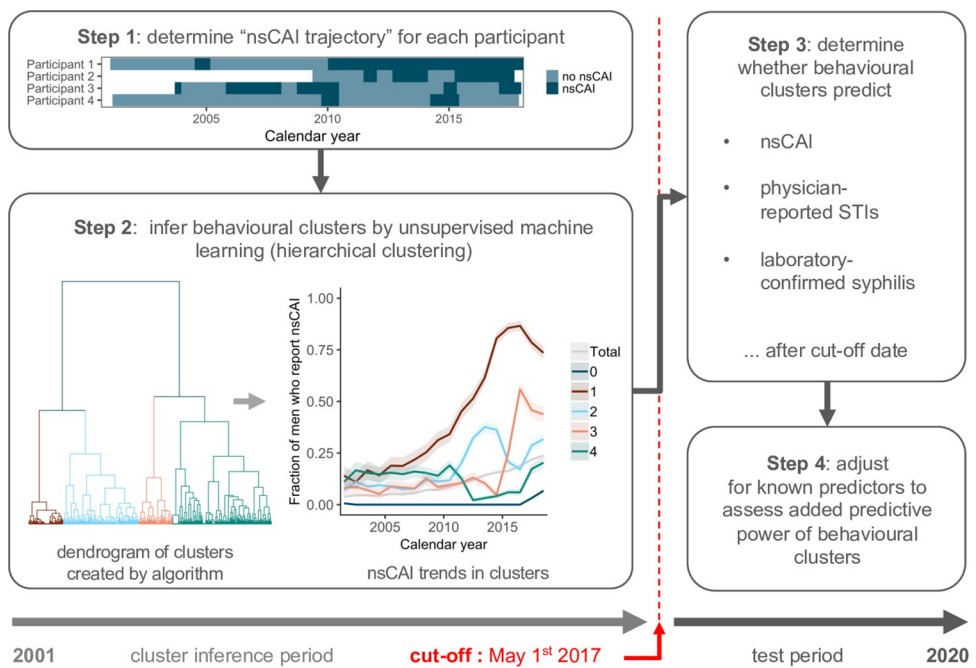

**Fig 1. Overview of clustering and prediction method.** nsCAI = condomless anal intercourse with non-steady partners.

proportion of discordant bits (the Jaccard distance) for each participant combination. Participants were then clustered by agglomerative hierarchical clustering using the R function *hclust* [20]. In this algorithm, each participant begins as a single cluster, and then pairs of clusters are successively merged in a way that dissimilarity within clusters is minimised (as computed by the Ward method [21]) until all participants are part of one single cluster (Fig 1, Step 2). Participants who never reported nsCAI were added as a separate cluster, cluster 0, and served as a baseline cluster.

All codes are available on https://www.github.com/Kouyos-Group/Behavioural-Clusters-and-STIs. For details see Section 1 in S1 Text.

## Behavioural clusters to predict sexual behaviour, STIs and syphilis

Besides the questions on sexual behaviour detailed above, the following variables are also recorded at SHCS follow-up and were defined as outcomes for our analyses:

a. *Nurse/physician-reported STIs*: The study nurse or physician interviewing the participant is asked to specify whether a participant had an STI since last follow-up (which may be diagnosed at the SHCS center or elsewhere), and to document the date of diagnosis of this STI. Nurse/physician-reported STIs may include—among others—syphilis, chlamydia or gonorrhea.

b. *Laboratory-confirmed syphilis*: Syphilis tests are routinely conducted in the SHCS at follow-up since 2004, consisting of a non-treponemal test for screening and a treponemal test for follow-up. An incident syphilis episode was defined as newly positive *Treponema pallidum* hemagglutinin or particle agglutination test or a venereal disease research laboratory (VDRL) titer of more than 1:8 and more than four-fold VDRL increase from the last VDRL titer.

As routine reporting of STIs by nurses and physicians started in late 2017 and the first recorded infection date for a nurse/physician-reported STI was early May 2017, we chose May 1st 2017 as our cut-off date. Behavioural clusters were thus inferred based on nsCAI reports from mid-2001 until May 1st 2017.

We used logistic regression to test the association between behavioural clusters and whether after May 1st 2017 (a) participants reported nsCAI at their first follow-up, (b) their physicians or study nurses reported an STI, or (c) incident syphilis was diagnosed in the laboratory (Fig 1, Step 3). We aimed to test whether behavioural clusters are predictive of nsCAI, STIs and syphilis, and hence chose non-overlapping timeframes to infer the clusters and compute the outcomes, respectively. To assess whether behavioural clusters were predictive beyond what could be inferred from known predictors for sexual behaviour and STIs, we adjusted our models for reported nsCAI in last follow-up before cut-off, syphilis prior to cut-off, and age (Fig 1, Step 4). We used likelihood ratio tests (LRT) and Bayesian information criteria (BIC) to compare model fits with and without behavioural clusters. Likelihood ratio tests were chosen as they can aptly compare the goodness of fit of nested models, while Bayesian information criteria were chosen for their over-/underfitting assessment properties. The main analyses were conducted without splitting the data into a test and validation dataset, as our analysis was not designed to be predictive in the classic sense. Further, we did not expect there to be a large risk of overfitting given the small number of parameters and the large number of events. We did however perform a receiver operator characteristic curve analysis, and repeated the analyses in a 5-fold cross-validation.

As another model evaluation, we compared behavioural clusters to other metrics derived from past behaviour: only the last nsCAI value available before cut-off, the last two available nsCAI values, an ever reported nsCAI and a mean nsCAI value over the whole study period before cut-off.

These analyses were repeated with clusters inferred based on sex with non-steady partners irrespective of condom use; and for predicting the number of nurse/physician-reported STIs and laboratory-confirmed syphilis episodes after cut-off.

## Behavioural clusters and number of sexual partners

A subset of the participants reported their number of sexual partners in the ZPHI at up to four time points per person between June 2015 and June 2016. We inferred clusters based on these participants' history of nsCAI from 2010 until June 2016, and used mixed effect Poisson and negative binomial regression to analyse whether behavioural clusters were associated with their number of sexual partners. Random effects were assumed for each person, while behavioural clusters, reported nsCAI and age were used as fixed effects. Models with and without behavioural clusters were compared using LRT and BIC.

We further analysed whether clusters were predictive of participants' number of sexual partners by inferring clusters with data exclusively until May 2015, and repeated all analyses analogously for clusters based on nsP.

## Varying number of behavioural clusters

Using hierarchical clustering to classify individuals yields a bifurcating tree. Each node in this tree indicates the division of a cluster into two clusters, and each tip represents one person. One can, in principle, choose an arbitrary number of clusters to consider. In the main analyses, we chose a medium numbers of clusters (five clusters to predict nsCAI, nurse/physician-reported STIs and laboratory-confirmed syphilis; and three to analyse the association with the number of sexual partners) in order to prevent having too few participants in a cluster on the

one hand, yet to retain narrative clarity on the other hand. However, as a sensitivity analysis, we repeated the analyses above with varying numbers of clusters. We compared these models using LRT and BIC.

### Behavioural cluster and SHCS center

Finally, we analysed whether participants in behavioural clusters clustered geographically using logistic regression.

## Results

6354 HIV-positive MSM were followed up in the SHCS between 2001 and 2020. Of these, 3710 had at least two years of follow-up with two or more nsCAI records between 2001 and 2017, and follow-up at least once after the cut-off date with a valid entry for each of the three outcomes considered. 122 participants additionally reported their number of sexual partners between 2015 and 2016 in the context of the ZPHI.

Out of 3710 participants, 828 (22%) reported nsCAI and 1659 (45%) reported nsP at their first follow-up after May 1$^{st}$ 2017, and between May 1$^{st}$ 2017 and the end of the study period in June 2020, 726 (20%) had a nurse/physician-reported STI, and 331 (9%) had incident laboratory-confirmed syphilis. Table 1 summarises characteristics of the participants included.

### Condom use patterns to predict future condom use, STIs and syphilis

We inferred four behavioural clusters among the included participants: Clusters 1 to 4 comprise 408 (11%), 628 (17%), 366 (10%) and 478 participants (13%), respectively. 1830 (49%) participants never reported nsCAI before cut-off and were added as an additional cluster, cluster 0. Fig 2 shows the nsCAI trends over time in the respective behavioural clusters.

In univariable models, behavioural clusters were significantly predictive of nsCAI, nurse/physician-reported STIs and laboratory-confirmed syphilis after cut-off (p-value of likelihood ratio test ($p_{LRT} < 0.001$ for all outcomes). Younger age, reporting nsCAI at last follow-up before cut-off and prior syphilis were also predictive of all outcomes (Figs A and B in S1 Text). In multivariable analysis adjusted for nsCAI before cut-off, prior syphilis and age, considering behavioural clusters in addition to these variables significantly improved the model fit for predicting nsCAI ($p_{LRT} < 0.001$) and nurse/physician-reported STIs ($p_{LRT} < 0.001$), as well as for predicting laboratory-confirmed syphilis ($p_{LRT} < 0.05$). When considering BIC, multivariable models with behavioural clusters performed better than those without clusters for predicting nsCAI (BIC = 2778 without vs. 2586 with clusters; Fig 3), but not for predicting nurse/physician-reported STIs and laboratory-confirmed syphilis (BIC = 3162 vs. 3164 for STIs and 2097 vs. 2120 for syphilis without and with clusters, respectively). Therefore, in multivariable analysis for predicting nsCAI, clusters improved model performance according to LRT and BIC, whereas for STI and syphilis incidence, clusters improved model performance according to LRT but not according to BIC.

Analysis of receiver operator characteristic curves showed consistently higher areas under the curve (AUC) for models including clusters vs. those without clusters, though the differences in AUC were small (for example AUC = 0.88 vs 0.84 for predicting nsCAI with vs. without clusters; Fig C in S1 Text). Repeating the analyses in a five-fold cross-validation yielded similar results. Accuracy for predicting future behaviour and STIs ranged from 78% to 91%, and adding the clusters to existing models brought only marginal benefit for most outcomes (Figs D and E in S1 Text).

Comparing models with behavioural clusters to models including other metrics derived from past behaviour showed that while clusters improve model fit, equal or better

**Table 1. Participant characteristics.** Mandatory schooling in Switzerland runs from grade one through grade nine. An apprenticeship would typically follow after these nine years and include three or more years of on-the-job training in combination with part-time classroom schooling.

| | Participants included in clusters to predict nsCAI/nsP, STIs and syphilis | Participants included in clusters to explain number of sexual partners (ZPHI subcohort) |
|---|---|---|
| Number of participants included | **N = 3710** | **N = 122** |
| *Follow-up period* | ***July 2001 to May 1st 2017*** | ***January 2010 to June 2016*** |
| Age at registration in the SHCS, median (IQR) | 37 (31–44) | 34 (29–41) |
| Year of registration in the SHCS, median (IQR) | 2006 (1998–2010) | 2009 (2007–2012) |
| Antiretroviral therapy ever started | 3676 (99%) | 122 (100%) |
| Viral RNA undetectable at end of follow-up | 3494 (94%) | 114 (93%) |
| Sex with non-steady partners during follow-up | 3189 (86%) | 114 (93%) |
| Condomless anal intercourse with non-steady partners during follow-up | 1880 (51%) | 99 (81%) |
| Recreational drug use during follow-up (excluding cannabis and alcohol) | 1156 (31%) | 49 (40%) |
| Education<br>  Mandatory school<br>  Finished apprenticeship<br>  Higher professional, technical or university education | 308 (8%)<br>1699 (46%)<br>1621 (44%) | 4 (3%)<br>74 (61%)<br>42 (34%) |
| European origin | 3318 (89%) | 113 (93%) |
| Laboratory-confirmed syphilis during follow-up | 1239 (33%) | 39 (32%) |
| More than one laboratory-confirmed episode of syphilis during follow-up | 418 (11%) | 12 (10%) |
| *Follow-up period* | ***May 2nd 2017 to June 2020*** | |
| Laboratory-confirmed syphilis after cut-off | 331 (9%) | - |
| More than one laboratory-confirmed syphilis episode after cut-off | 198 (5%) | - |
| Nurse/physician-reported STI after cut-off | 726 (20%) | - |
| More than one nurse/physician-reported STI after cut-off | 290 (8%) | - |

improvements can be achieved by considering other parameters, such as the last two available nsCAI values, or using a mean nsCAI value before cut-off. Models considering behavioural clusters performed consistently better than those only considering whether a participant had ever reported nsCAI (Fig F in S1 Text).

Table 2 summarises the extent to which clusters improve prediction of nsCAI, STIs and syphilis.

## nsCAI trends in behavioural clusters

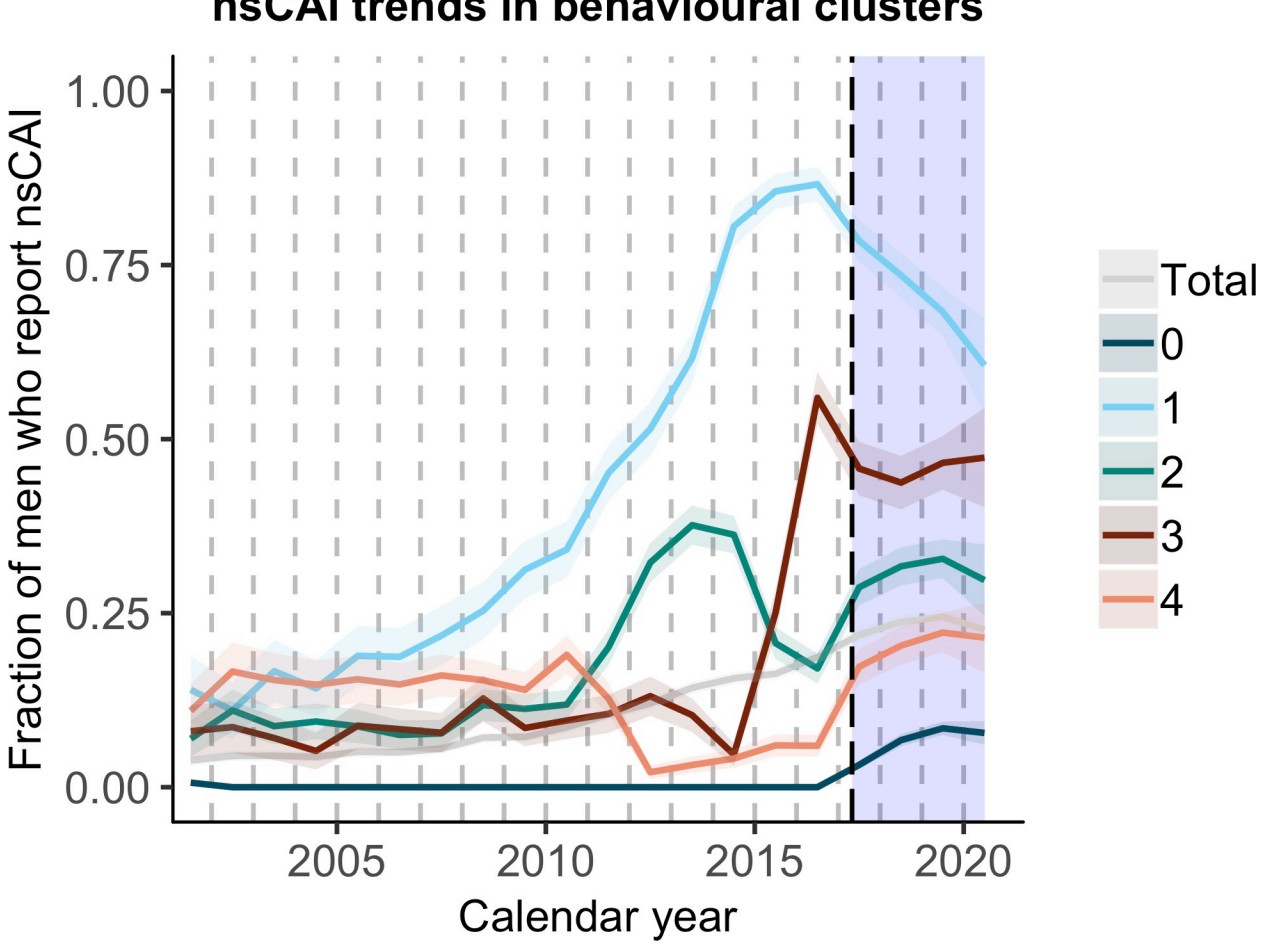

**Fig 2. Trends in nsCAI proportion in behavioural clusters.** The clusters contain the following proportions of the study population (from 0 through 4): 49%, 11%, 17%, 10% and 13%. Cluster 0 consists of participants who never reported nsCAI during the observation period (in this case mid-2001 until the cut-off date). The dashed vertical line on the 1st of May 2017 represents the cut-off date between the observation period (which corresponds to the period used to infer the clusters) and the outcome period. The shaded area shows the time in which nsCAI, nurse/physician-reported STIs and laboratory-confirmed syphilis were recorded as outcomes. nsCAI = condomless anal intercourse with non-steady partners.

### Patterns of non-steady partners to predict future non-steady partners, STIs and syphilis

Behavioural clusters based on nsP (Fig G in S1 Text) were significantly predictive for future nsP, nurse/physician-reported STIs and laboratory-confirmed syphilis after cut-off in univariable analysis (Fig H in S1 Text). Moreover, adding behavioural clusters to models considering nsP before cut-off, prior syphilis and age significantly improved model fit for all outcomes (Table 2). According to BIC, the model with behavioural clusters showed better performance for prediction of nsP, but not for nurse/physician-reported STIs and laboratory-confirmed syphilis (Fig I in S1 Text).

### Patterns in condom use and non-steady partners to predict *number* of future STIs and syphilis episodes

290 participants (8%) had more than one nurse/physician-reported STI and 198 (5%) had more than one laboratory-confirmed syphilis episode after cut-off. In multivariable Poisson

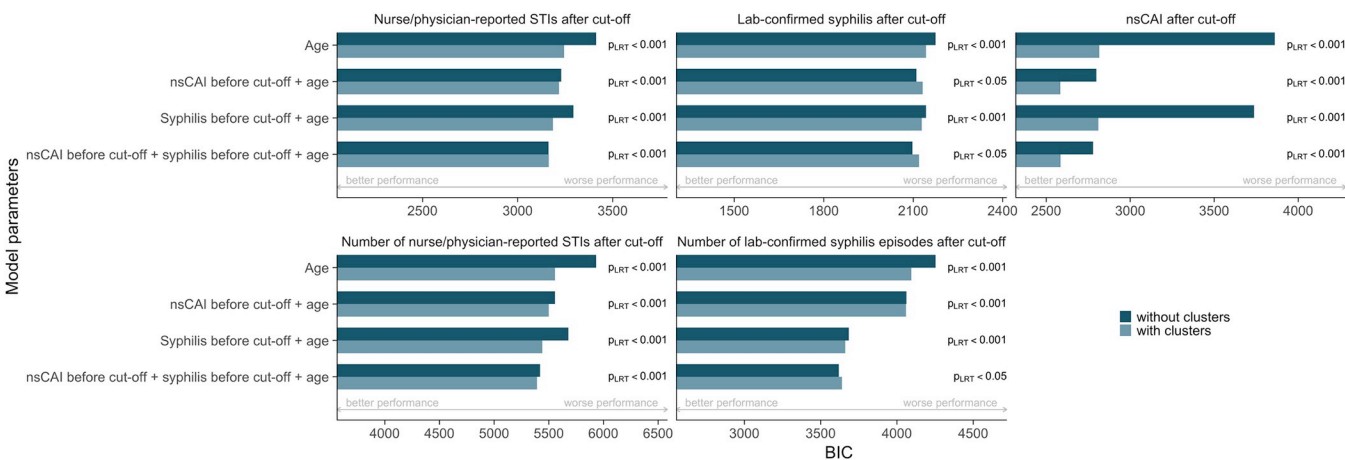

**Fig 3. Bar plots for Bayesian information criterion values.** Regression models with different combinations of predictor variables to predict nurse/physician-reported STIs (left), laboratory-confirmed syphilis (center) and nsCAI (right) after cut-off. A smaller BIC represents a better prediction. Numbers to the right of the bars represent the p value of the likelihood ratio test between the two models in question. nsCAI = condomless anal intercourse with non-steady partners. BIC = Bayesian information criterion. $p_{LRT}$ = p value of likelihood ratio test comparing the respective models with and without clusters.

regression, adding clusters to models considering nsCAI before cut-off, number of prior syphilis episodes and age improved model performance according to both LRT and BIC for predicting the number of nurse/physician-reported STIs ($p_{LRT}$ < 0.001, BIC = 5419 without vs. 5391 with clusters. Fig 3), and according to LRT but not BIC for predicting the number of

**Table 2. Summary of model comparison for predictions.** Columns from left to right: (1) Outcomes. nsCAI = condomless anal intercourse with non-steady partners, STI = sexually transmitted infection, nsP = sex with non-steady partners. (2) P value of likelihood ratio test ($p_{LRT}$) of univariable model with behavioural clusters vs. null model. (3) $p_{LRT}$ of multivariable model including age, last nsCAI or nsP status, prior syphilis (or number of prior syphilis episodes for count outcomes) as predictor variables vs. the same model plus behavioural clusters. (4 through 9) Akaike information criteria (AIC), Bayesian information criteria (BIC), and area under the receiver operator characteristic curve (auROC) for multivariable model including age, last nsCAI or nsP status, prior syphilis (or number of prior syphilis episodes for count outcomes) as predictor variables and of the same model plus behavioural clusters. Clusters included in the models were inferred based on long-term nsCAI patterns for the top half of the table or long-term nsP patterns for the bottom half of the table. Green shading means a better performance for the model with clusters.

| Outcome | $p_{LRT}$ null model vs. clusters | $p_{LRT}$ without vs. with clusters | AIC | | BIC | | auROC | |
|---|---|---|---|---|---|---|---|---|
| | | | without clusters | with clusters | without clusters | with clusters | without clusters | with clusters |
| *Clusters based on nsCAI patterns* | | | | | | | | |
| nsCAI | <0.001 | <0.001 | 2754 | 2536 | 2778 | 2586 | 0.84 | 0.88 |
| Any nurse/physician-reported STI | <0.001 | <0.001 | 3137 | 3114 | 3162 | 3164 | 0.77 | 0.77 |
| Number of nurse/physician-reported STIs | <0.001 | <0.001 | 5394 | 5341 | 5419 | 5391 | 0.79 | 0.80 |
| Laboratory-confirmed syphilis | <0.001 | <0.05 | 2072 | 2070 | 2097 | 2120 | 0.71 | 0.72 |
| Number of laboratory-confirmed syphilis episodes | <0.001 | <0.05 | 3594 | 3589 | 3619 | 3639 | 0.72 | 0.78 |
| *Clusters based on nsP patterns* | | | | | | | | |
| nsP | <0.001 | <0.001 | 3351 | 3168 | 3376 | 3218 | 0.85 | 0.88 |
| Any nurse/physician-reported STI | <0.001 | <0.001 | 3092 | 3080 | 3117 | 3130 | 0.78 | 0.78 |
| Number of nurse/physician-reported STIs | <0.001 | <0.001 | 5313 | 5259 | 5338 | 5309 | 0.81 | 0.82 |
| Laboratory-confirmed syphilis | <0.001 | <0.05 | 2062 | 2057 | 2087 | 2106 | 0.72 | 0.73 |
| Number of laboratory-confirmed syphilis episodes | <0.001 | <0.001 | 3562 | 3536 | 3587 | 3586 | 0.78 | 0.79 |

laboratory-confirmed syphilis episodes ($p_{LRT} < 0.05$, BIC = 3619 without vs. 3639 with clusters. Fig 3).

Clusters based on nsP patterns improved predictions by known risk factors for all outcomes according to pLRT, AIC and auROC, though not according for BIC for nurse/physician-reported STIs and syphilis (Table 2).

### Association of behavioural clusters with number of sexual partners

Among the 122 participants additionally enrolled in the ZPHI, we inferred a set of three behavioural clusters: 23 participants (19%) never reported nsCAI during follow-up and served as the reference cluster, cluster 0. Clusters 1 and 2 were inferred using hierarchical clustering and comprise 29 (24%) and 70 participants (57%), respectively. Fig J in S1 Text shows the nsCAI trends in the respective clusters.

The median number of sexual partners in the three months preceding follow-up was 2 (IQR = 1–5). Members of cluster 0 reported the fewest sexual partners with a median of 1 (IQR = 1–2), while those in clusters 1 reported the most, with a median of 4 (IQR = 3–10). Members of cluster 2 reported a median of 2 partners (IQR = 1–4). The behavioural clusters showed significant differences in number of sexual partners in univariable models (Fig K in S1 Text), and considering behavioural clusters in addition to reported nsCAI and age improved model fit ($p_{LRT} < 0.01$, BIC = 1150 without vs. 1147 with clusters). Similar results were obtained using negative binomial models. Clusters based on nsP patterns were also significantly associated with the number of sexual partners, however this effect was no longer seen after adjusting for nsP and age ($p_{LRT} = 0.38$, BIC = 1135 without vs. 1144 with clusters).

A set of three behavioural clusters based on nsCAI patterns was predictive of participants' number of sexual partners after cut-off in univariable models and after adjusting for nsCAI and age ($p_{LRT} < 0.05$). Clusters based on nsP were predictive of participants' number of partners in univariable models, though not when adjusting for nsP and age ($p_{LRT} = 0.16$).

### Varying number of clusters

The patterns seen in the primary analyses were robust when varying the number of clusters: according to BIC, a range of behavioural clusters enhanced predictions of nsCAI and nurse/physician-reported STIs. However, no number of clusters improved model performance for predicting syphilis (Fig L in S1 Text). Qualitatively similar results were observed for clusters based on nsP patterns (Fig M in S1 Text). Table 3 summarises the ideal number of clusters according to BIC for predicting the outcomes studied.

### Behavioural clusters and SHCS center

Clusters were significantly associated with SHCS center ($p_{LRT}$ vs. null model $< 0.05$).

### Discussion

Our results show that behavioural clusters inferred by unsupervised machine learning can improve predictions of future sexual behaviour and STI acquisition, and are significantly associated with individuals' number of sexual partners. By clustering participants based solely on their history of sexual behaviour, this approach provides an alternative to potentially simplifying categorisations by demographic or static variables.

Behavioral clusters improved prediction of STI acquisition beyond information available from routinely collected data in the SHCS. This may be because individuals in the same cluster also share other aspects of sexual behaviour that are not recorded in most longitudinal cohorts,

**Table 3. Ideal number of behaviour clusters to enhance predictions according to BIC.** A dash means adding behavioural clusters to prediction models did not improve BIC. BIC = Bayesian information criterion. nsCAI = condomless anal intercourse with non-steady partners. nsP = sex with non-steady partners.

| Outcome | Ideal number of clusters according to BIC |
|---|---|
| *Clusters based on nsCAI patterns* | |
| nsCAI | 5 |
| Any nurse/physician-reported STI | 4 |
| Number of nurse/physician-reported STIs | 5 |
| Laboratory-confirmed syphilis | - |
| Number of laboratory-confirmed syphilis episodes | - |
| *Clusters based on nsP patterns* | |
| nsP | 10 |
| Any nurse/physician-reported STI | 3 |
| Number of nurse/physician-reported STIs | 3 |
| Laboratory-confirmed syphilis | - |
| Number of laboratory-confirmed syphilis episodes | 3 |

such as sexualised drug use or group sex. The finding that clusters are associated with participants' number of sexual partners supports this idea. This hypothesis can be further explored in qualitative studies or quantitative studies with more detailed data on sexual practices. Another explanation could be that clustering individuals based on longitudinal behaviour data identifies sexual contact networks, i.e. that individuals with similar behaviour over time may be more likely to have sex together, thus explaining similar STI incidence [22]. One could assume that individuals in the same sexual contact network cluster geographically and are thus registered at the same SHCS center. An exploratory analysis of the association of SHCS center and STI was suggestive of this. This hypothesis can be more adequately tested using phylogenetic data, as was explored in a recent proof of concept study using HCV phylogenies [23].

Our study has strengths and limitations. A key strength of the study is the rich longitudinal and representative data collected in the SHCS and ZPHI: This analysis draws on a population of nearly four thousand HIV-positive MSM who were asked the same questions on sexual behaviour consistently over a period of twenty years. A limitation is that the term "non-steady partner" is not further defined, thus representing a source of uncertainty and potential bias. Similarly, nurse/physician-reported STIs, an SHCS variable introduced in 2017, are reported by the interviewer at follow-up and may thus be subject to desirability or other biases, and asymptomatic STIs are likely to go undetected. By contrast, syphilis is routinely screened for at follow-up since 2004. We decided to include both STI variables as outcomes as we believed this showed a more complete picture of the association between the clusters and STIs. As the nurse/physician-reported STIs are only recorded since 2017, we adjust our models only for prior syphilis and not for prior nurse/physician-reported STIs. Though a limitation, this condition represents a scenario in which behavioural clusters could be valuable: to complement available information where data on the outcome of interest is absent. It is conceivable that the stronger association between cluster membership and nurse/physician-reported STIs compared to laboratory-confirmed syphilis is mediated by reporting bias (i.e. participants reporting more condomless sex being systematically more likely to also self-report STIs). Additionally, individuals with similar sexual behaviour over time may be more likely to share similar STI testing habits, which in turn affects the incidence of self-reported STIs but not of laboratory-confirmed syphilis, as the latter is part of routine screening in the SHCS.

To allow for a clearer presentation of the complex associations and trends in sexual behaviour in the SHCS, we restricted our analysis to two behavioural variables: condomless anal

intercourse with non-steady partners and sex with non-steady partners. Condomless anal intercourse with non-steady partners has repeatedly been shown to be a meaningful STI exposure variable recorded in the SHCS and ZPHI [7,11,24]. Nevertheless, we found similar results for clusters based on condomless anal intercourse with non-steady partners and clusters based on sex with non-steady partners irrespective of condom use. This is consistent with previous findings that non-steady partners are a risk factor for syphilis whether condoms are used or not [8].

Our analyses are based solely on MSM in the SHCS—a predominantly European, highly educated, HIV-positive population with nearly 100% ART coverage. Our findings may thus not be generalizable to HIV-negative MSM. However, we believe that the MSM population in the SHCS is highly representative of HIV-positive MSM in Switzerland [18], and as such we believe to some extent representative of HIV-positive MSM populations in other high-income European settings.

To our knowledge, this is the first study to assess STI predictions based on subgroups inferred by hierarchical clustering. Several studies have used latent class analysis to disentangle patterns in sexual behaviour [25–30], and in many instances, have found differences in STI incidence between classes. For example, Achterbergh et al. found that clustering MSM in Amsterdam and surrounding regions based on drug use patterns revealed groups with different numbers of sexual partners and STI prevalence [26]. Our study differs from existing publications in three main aspects: First, we chose to create clusters using hierarchical clustering. In contrast to latent class analysis, hierarchical clustering yields a full representation of the cluster structure in which each individual is fully traceable, and does not involve hypotheses regarding the underlying data structure. Second, in contrast to other longitudinal studies, we consider calendar time rather than patient time, allowing the clusters to capture effects of historical events such as the diffusion of the Swiss Statement in 2008 [31] and the "undetectable = untransmissible" message in subsequent years [16,32,33]. Finally, existing studies normally consider concurrent STI incidence. The longitudinal nature of the SHCS allowed us to infer clusters using data until a certain cut-off date, and compute outcomes after this date, showing that clusters can be useful for predicting future events.

We recognise that there may be more performant ways to predict STIs. For instance, we found that while behavioural clusters do improve predictions of sexual behaviour and STIs, equal or better results can be achieved considering the last two records on sexual behaviour, or a mean thereof over the whole study period. Further, the improvements in model fit were less pronounced when using a validation data set. However, the aim of this study was to test the relevance of behavioural clusters for understanding the epidemiology of STIs (i.e. to assess whether these clusters were associated with distinct patterns of STI incidence), rather than to find the ideal way to predict STIs within the SHCS.

We test the association between behavioural clusters and STIs in a predictive context but, more generally, this analysis also informs about which dimensions of human behaviour matter most for STI incidence and how complex temporal variation of behavioural data can be best simplified to capture these essential dimensions.

The clustering and prediction framework presented here can be adapted to any type of longitudinal data. Hierarchical clustering allows a large flexibility in the metric chosen to determine distance between pairs of samples: While we used single binary features over time here (condomless anal intercourse with non-steady partners or sex with non-steady partners), the algorithm can easily accommodate multiple features as well as continuous or discrete values with more than two levels.

In summary, we find that clustering individuals based on their sexual behaviour over time using unsupervised machine learning identifies subgroups of the MSM population in the

SHCS with distinct STI exposure. Our findings have implications for research on sexual health and STIs in that we propose an alternative method for creating exposure categories for infectious disease modelling. Furthermore, this study contributes to the evidence on machine learning applications in epidemiology by validating a flexible unsupervised machine learning framework adaptable to any type of longitudinal data.

## Supporting information

**S1 Text. Supplementary Material.**
(PDF)

## Acknowledgments

We thank the patients for participating in the Swiss HIV Cohort Study (SHCS), the study nurses, physicians, data managers, and the administrative assistants. Members of the SHCS: Abela I, Aebi-Popp K, Anagnostopoulos A, Battegay M, Bernasconi E, Braun DL, Bucher HC, Calmy A, Cavassini M, Ciuffi A, Dollenmaier G, Egger M, Elzi L, Fehr J, Fellay J, Furrer H, Fux CA, Günthard HF (President of the SHCS), Hachfeld A, Haerry D (deputy of "Positive Council"), Hasse B, Hirsch HH, Hoffmann M, Hösli I, Huber M, Jackson-Perry D (patient representatives), Kahlert CR (Chairman of the Mother & Child Substudy), Kaiser L, Keiser O, Klimkait T, Kouyos RD, Kovari H, Kusejko K (Head of Data Centre), Labhardt N, Leuzinger K, Martinez de Tejada B, Marzolini C, Metzner KJ, Müller N, Nemeth J, Nicca D, Notter J, Paioni P, Pantaleo G, Perreau M, Rauch A (Chairman of the Scientific Board), Salazar-Vizcaya L, Schmid P, Speck R, Stöckle M (Chairman of the Clinical and Laboratory Committee), Tarr P, Trkola A, Wandeler G, Weisser M, Yerly S.

## Author Contributions

**Conceptualization:** Sara Andresen, Suraj Balakrishna, Axel J. Schmidt, Dominique L. Braun, Huldrych F. Günthard, Andri Rauch, Roger D. Kouyos, Luisa Salazar-Vizcaya.

**Data curation:** Sara Andresen, Suraj Balakrishna, Catrina Mugglin, Dominique L. Braun.

**Formal analysis:** Sara Andresen, Luisa Salazar-Vizcaya.

**Methodology:** Sara Andresen, Axel J. Schmidt, Huldrych F. Günthard, Andri Rauch, Roger D. Kouyos, Luisa Salazar-Vizcaya.

**Resources:** Dominique L. Braun, Thanh Doco Lecompte, Katharine EA Darling, Jan A. Roth, Patrick Schmid, Enos Bernasconi, Huldrych F. Günthard, Andri Rauch, Roger D. Kouyos.

**Software:** Sara Andresen, Luisa Salazar-Vizcaya.

**Supervision:** Suraj Balakrishna, Andri Rauch, Roger D. Kouyos, Luisa Salazar-Vizcaya.

**Visualization:** Sara Andresen.

**Writing – original draft:** Sara Andresen.

**Writing – review & editing:** Sara Andresen, Suraj Balakrishna, Catrina Mugglin, Axel J. Schmidt, Dominique L. Braun, Alex Marzel, Thanh Doco Lecompte, Katharine EA Darling, Jan A. Roth, Patrick Schmid, Enos Bernasconi, Huldrych F. Günthard, Andri Rauch, Roger D. Kouyos, Luisa Salazar-Vizcaya.

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
