## [Decision Letter · Decision Letter 0]

12 Apr 2022

Dear Mrs. Andresen,

Thank you very much for submitting your manuscript "Unsupervised machine learning predicts future sexual behaviour and sexually transmitted infections among HIV-positive men who have sex with men" for consideration at PLOS Computational Biology.

As with all papers reviewed by the journal, your manuscript was reviewed by members of the editorial board and by several independent reviewers. In light of the reviews (below this email), we would like to invite the resubmission of a significantly-revised version that takes into account the reviewers' comments.

Both Reviewers saw this investigation as worthwhile. The major outstanding issue with this manuscript is whether "training" and "validation" datasets were employed (per Reviewer 1). Upon re-reading this manuscript, I agree that is not clear. Though a charitable reading should suggest that "up to a certain cut-off date, and apply regression to test" indicates this approach. However, if this is the case, its presentation is too informal. I recommend the authors clarify their approach and perform a more robust assessment of their model, as re-review will be necessary before any formal consideration of this manuscript.

We cannot make any decision about publication until we have seen the revised manuscript and your response to the reviewers' comments. Your revised manuscript is also likely to be sent to reviewers for further evaluation.

Sincerely,

Joel O. Wertheim

Associate Editor

PLOS Computational Biology

Thomas Leitner

Deputy Editor

PLOS Computational Biology

Both Reviewers saw this investigation as worthwhile. The major outstanding issue with this manuscript is whether "training" and "validation" datasets were employed (per Reviewer 1). Upon re-reading this manuscript, I agree that is not clear. Though a charitable reading should suggest that "up to a certain cut-off date, and apply regression to test" indicates this approach. However, if this is the case, its presentation is too informal. I recommend the authors clarify their approach and perform a more robust assessment of their model, as re-review will be necessary before any formal consideration of this manuscript.

Reviewer's Responses to Questions

**Comments to the Authors:**

Reviewer #1: The manuscript entitled “Unsupervised machine learning predicts future sexual behaviour and sexually transmitted infections among HIV-positive men who have sex with men” used statistical approaches to assess whether clusters (developed using a hierarchical clustering technique) enhance predictions of sexual behaviour or sexually transmitted diseases (STIs).

Overall, I thought the paper read nicely and addressed an important policy question. However, I thought there were some methodological gaps. Below are the gaps that I think should be addressed. Thank you for letting me read your manuscript!

Major comments

1. The authors used likelihood ratio test (LRT), AIC, BIC, and auROC to assess the predictive performance of the clusters. To my knowledge, LRT, AIC, and BIC are typically used for model selection and not prediction. While these three metrics showed that a model including the cluster variables improves model fit, these metrics do not assess prediction. Therefore, based on my understanding, statements such as those on lines 174 (“…improved the model fit for predicting”) and 180 (“…improved model performance”) are not accurate as LRT does not assess prediction.

2. The manuscript did not discuss the use of a training and validation datasets. As written, it seems that the entire dataset was used to assess prediction. Without the use of training/validation datasets the prediction is typically too optimistic. Creating training/validation datasets seem to be the standard approach for prediction, so it would be nice to understand why that was not used.

3. The paper makes that claim that the use of the clusters increases the prediction. However, it would be nice to see a more robust model selection framework, i.e., including the previous two nsCAI values as variables (instead of just the previous one) or an “ever nsCAI” variable as well as investigation of the functional form of age.

4. The auROC metric (which was the one metric in the paper that I typically have seen used to assess prediction) did not seem very different with and without the clusters (as seen in Table 2). This left me wondering if the unsupervised machine learning clusters really did increase prediction. Especially once training and validation datasets are created and a more robust model selection approach is taken.

Minor comments

1. Not sure if “corrected our models” is the right term; maybe use control or adjust (see line 108 for an example).

2. I was not clear on lines 281-284 regarding the sexual contact network. It would be helpful to include more details on the connection between the analysis and contact networks.

Reviewer #2: Thank you for giving me the opportunity to review this manuscript. I thought it was well written and the subject is interesting. Although the subject of machine learning is not my expertise, I do have some minor points that will hopefully help to further improve manuscript.

Abstract

- 2nd paragraph: I find “up to a certain cut-off point” rather vague. I would suggest just to provide the cut-off date here.

- I am not sure what the author mean with the last paragraph of the abstract and I do not think this is elaborated on the Discussion section of the main paper. Do you mean the clusters with “framework”? And how can this framework be used as an alternate method for categorization (this is also mentioned in the Conclusion of the main paper) and how can it contribute to a better understanding of time-varying risk factors?

Methods:

- Line 59: how did you define a non-steady partner? Or a steady partner?

- Line 65: suggestion to refrain from using abbreviations that are not commonly used such as “nsCAI” and “nsP” if the word count allows it. This would increase the readability of the manuscript.

Results:

- Figure 1: panels for step 1 and 2 are rather small now and the text was very hard to read. Would is be possible to increase the size?

- Why was only syphilis routinely tested and not chlamydia and gonorrhea?

- Table 1: could you explain what is considered “mandatory schooling” in Switzerland? Is that similar to primary school and high school for example? And what school level corresponds to “finished apprenticeship?”

- Figure 2: I am not sure if the “total” line is necessary in this figure

Discussion:

- Line 284: do you mean the sensitivity analysis last mentioned in the Results section here? If so, I would clarify this.

- Line 292: Indeed, asymptomatic STIs may likely go unnoticed and thereby impact your outcome if it is based on self-report. How do you think this would impact your results if, instead of self-report, the outcome was based on actual STI testing done at follow-up visits?

**Have the authors made all data and (if applicable) computational code underlying the findings in their manuscript fully available?**

Reviewer #1: Yes

Reviewer #2: **No: **see reasons in manuscript

PLOS authors have the option to publish the peer review history of their article (what does this mean?). If published, this will include your full peer review and any attached files.

Reviewer #1: No

Reviewer #2: No
---

## [Editor Report · Decision Letter 1]

18 Jul 2022

Dear Mrs. Andresen,

Thank you very much for submitting your manuscript "Unsupervised machine learning predicts future sexual behaviour and sexually transmitted infections among HIV-positive men who have sex with men" for consideration at PLOS Computational Biology. As with all papers reviewed by the journal, your manuscript was reviewed by members of the editorial board. The Editor appreciated the attention to an important topic, and we are likely to accept this manuscript for publication, providing that you modify the manuscript according to the below recommendations.

Having reviewed the response to reviewers and revised manuscript, I believe the authors have appropriately responded to the reviewers comments and criticism. However, these responses are not yet sufficiently reflected in the main text of the manuscript. Rather, many legitimate points of confusion and ambiguity were addressed only in the Response to Reviewers document, and not within the text to be published itself. This approach leaves open the strong possibility that readers of this paper will be equally as confused as the reviewers. Therefore, I encourage the authors to revise their manuscript again to ensure that the accessibility of this manuscript. Please return to the original reviewers and go through point-by-point to ensure all important points are addressed in the manuscript itself. Further, I would make it clear why no validation set was included, as readers will as themselves have the same question, and your response will improve the impact of this study.

Sincerely,

Joel O. Wertheim

Associate Editor

PLOS Computational Biology

Thomas Leitner

Deputy Editor

PLOS Computational Biology

[LINK]

Having reviewed the response to reviewers and revised manuscript, I believe the authors have appropriately responded to the reviewers comments and criticism. However, these responses are not yet sufficiently reflected in the main text of the manuscript. Rather, many legitimate points of confusion and ambiguity were addressed only in the Response to Reviewers document, and not within the text to be published itself. This approach leaves open the strong possibility that readers of this paper will be equally as confused as the reviewers. Therefore, I encourage the authors to revise their manuscript again to ensure that the accessibility of this manuscript. Please return to the origin reviewers and go through point-by-point to ensure all important points are addressed in the manuscript itself. Further, I would make it clear why no validation set was included, as readers will as themselves the same question, and your response will improve the impact of this study.

Figure Files:

Data Requirements:

Reproducibility:

References:

---

## [Editor Report · Decision Letter 2]

12 Sep 2022

Dear Mrs. Andresen,

We are pleased to inform you that your manuscript 'Unsupervised machine learning predicts future sexual behaviour and sexually transmitted infections among HIV-positive men who have sex with men' has been provisionally accepted for publication in PLOS Computational Biology.

Best regards,

Joel O. Wertheim

Academic Editor

PLOS Computational Biology

Thomas Leitner

Section Editor

PLOS Computational Biology

---

## [Editor Report · Acceptance letter]

24 Oct 2022

PCOMPBIOL-D-21-02210R2 

Unsupervised machine learning predicts future sexual behaviour and sexually transmitted infections among HIV-positive men who have sex with men

Dear Dr Andresen,

I am pleased to inform you that your manuscript has been formally accepted for publication in PLOS Computational Biology. Your manuscript is now with our production department and you will be notified of the publication date in due course.

With kind regards,

Zsofia Freund
